# ACCESS: Prompt Engineering for Automated Web Accessibility Violation Corrections

## Abstract

With the increasing need for inclusive and user-friendly technology, web accessibility is crucial to ensuring equal access to online content for individuals with disabilities, including visual, auditory, cognitive, or motor impairments. Despite the existence of accessibility guidelines and standards such as Web Content Accessibility Guidelines (WCAG) and the Web Accessibility Initiative (W3C), over 90% of websites still fail to meet the necessary accessibility requirements. For web users with disabilities, there exists a need for a tool to automatically fix web page accessibility errors. While research has demonstrated methods to find and target accessibility errors, no research has focused on effectively correcting such violations. This paper presents a novel approach to correcting accessibility violations on the web by modifying the document object model (DOM) in real time with foundation models. Leveraging accessibility error information, large language models (LLMs), and prompt engineering techniques, we achieved greater than a 51% reduction in accessibility violation errors after corrections on our novel benchmark: ACCESS. Our work demonstrates a valuable approach toward the direction of inclusive web content, and provides directions for future research to explore advanced methods to automate web accessibility.

## 1 Introduction

Web accessibility refers to the extent to which online information is readily available to people, regardless of their physical or cognitive capabilities and the technical specifications of the devices they use (Wu et al., 2017). When web content, such as images, language, or display is not transmitted in a comprehensible manner, a web accessibility barrier is formed between the information and its receiver (Abu Addous et al., 2016). Particularly for websites largely dependent on graphical or multidimensional presentation formats, many individuals with impairments or disabilities are unable to effectively perform practical tasks such as navigating the website or completing forms (Kottapally et al., 2002). As a result, laws such as the Web Content Accessibility Guidelines (WCAG) have been established as international standards for enforcing accessibility. The WCAG follows thirteen guidelines to ensure accessible websites, organized under four principles: 1) perceivable, 2) operable, 3) understandable, and 4) robust (Wille et al., 2016).

Despite the enactment of laws aimed at promoting web accessibility, numerous websites still fail to meet the necessary accessibility standards. A 2023 study on the highest trafficked one million home pages found that each site presented 50 accessibility violations on average. Furthermore, the study found that 96.3% of all home pages had WCAG 2.0 failures, including the 22.1% of home pages that had missing alternate text for images (webaim, 2023). This issue can be attributed to the expensive, time-consuming, error-prone, and labor-intensive nature of manually detecting and individually addressing accessibility violations. Moreover, it is time-consuming for the Cognitive and Learning Disabilities Accessibility Task Force (COGA) to implement new accessibility guidelines, as issues surrounding the ability to clearly, objectively, and universally verify the implementation of guidelines can be an obstacle to their acceptance (Lewis & Seeman, 2019). Unless assistive technologies are designed in a way to address the need for interpretation by users, barriers to accessibility will continue to exist (Craven, 2006).

In recent years, research has suggested new possibilities for addressing web accessibility violations in a less resource-intensive manner, offering promising solutions to help developers detect viola-

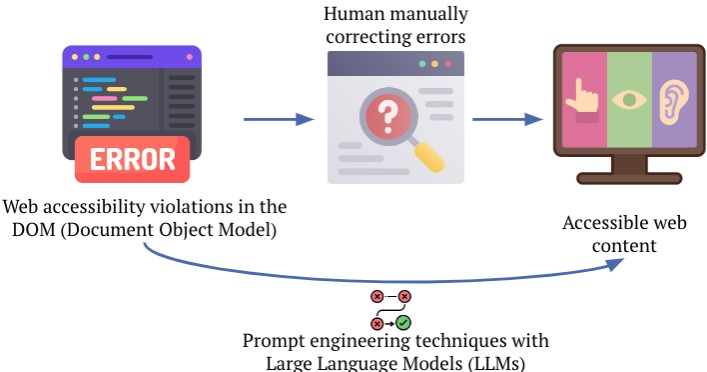

Figure 1: The big picture approach of our research. By automating accessibility violation corrections, we aim to eliminate the inefficiency and inaccuracy associated with manual error correction.

tions. Early research in Assistive Technology (AT) have been widely integrated into the validation process; however, early AT tools have shown to function poorly when used on dynamic content and web pages (Petrie et al., 2008). More recently, in order to efficiently and accurately identify web violations of a website according to WCAG standards, web evaluation tools standards such as Playwright, Tenon, and AChecker have been created. Most evaluation tools work by performing an automatic test for potential accessibility issues by traversing web page's DOM and performing complex rule-based checks in accordance to regulation standards. These tools are critical in ensuring compliance with accessibility requirements, and since a combination of multiple tools is often used to increase the percentage of detected errors. There exists a broad field of study for the development of web error detection tools, especially through the application of advanced programming techniques derived from artificial intelligence (Tiwary & Mahapatra, 2022). However, it remains important to note that violation detection tools can only help developers who actively seek to ensure compliance with accessibility requirements. Individuals with disabilities still cannot fully access the vast majority of web pages that fail to meet compliance requirements.

Consequently, by leveraging data gathered from these evaluation tools, recent research has emphasized the potential of automatically improving accessibility. For example, researchers have explored computer vision techniques to automatically correct alternate-text errors (Draffan et al., 2020). However, the flaws of current automatic strategies are twofold. First, their scope in current research is limited to the generation of alternate text for images. Furthermore, fully automatic strategies can have flaws in accuracy, especially with the lack of research in adaptive web accessibility (Baptista et al., 2016). Research examining Facebook's automatic alt-text feature demonstrated that although it increased user engagement, the automated alt-text was not significantly accurate or reliable, failing to understand objects in context. For example, Facebook's automatic alt-text approach was shown to overemphasize body parts and beards, while neglecting other parts of the image, leaving users confused about the true relevance of sections of the image (Abou-Zahra et al., 2018), (Yao et al., 2023).

Despite the promise of automatic accessibility violation corrections for alt-text generation, studies have undermined their functionality due to their shortcomings. Rather than a fully automatic approach, a semi-automatic approach has been explored, using similar automatic computer vision algorithms first but following with manual testing and improvements to increase accuracy (Adam & Kreps, 2006). For example, Tiwary and Mahapatra's semi-automatic algorithm used computer vision techniques followed by manual recommendations, achieving higher accuracy when compared to the human-written alt text, effectively striking a balance between detection and accuracy in correction (Wille et al., 2016). Emerging research has also introduced a representation learning approach for web pages, which addresses the computational expense, time, and lack of useful information limitations from previous research by utilizing a transformer-based encoder to learn generalizable representations of HTML documents via self-supervised learning. The proposed approach outperforms baseline models designed for these tasks, suggesting the attractiveness of a natural language

Figure 2: Three example visual website violations that could affect web accessibility for individuals with impairments. **(a)** Contrast between foreground and background colors do not meet WCAG 2AA minimum contrast ratio thresholds. **(b)** Button does not have discernable text. **(c)** All page content is not contained by landmarks.

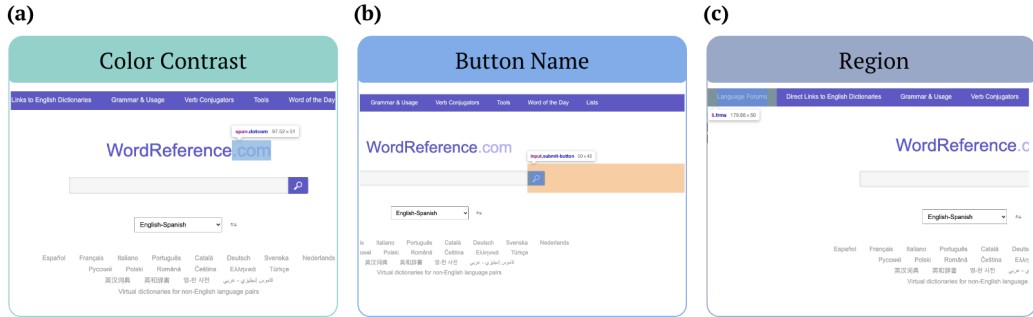

processing (NLP) approach to handling HTML documents and its utility toward real-world web applications (Deng et al., 2022).

The goal of our project is to create a novel automatic web accessibility correction tool for individuals with impairments. Through the use of prompt engineering, we can reap the inherent cost-efficient nature of automatic approaches while gaining the accuracy of semi-automatic approaches through prompt engineering. Currently, no accessibility correction tool exists with the intent of serving users with a tool to correct web pages on-the-go. While the scope of previous studies was limited to addressing alt-text generation with computer vision, our tool can be used more ubiquitously and purposefully by web users to handle web accessibility violations when desired. Our approach utilizes the Playwright API and a data set of website URLs to perform web accessibility tests to explore the viability of machine learning algorithms in addressing the limitations of previous web accessibility research in an efficient manner. Leveraging prompt engineering techniques, we aim to improve the accuracy of web accessibility violation corrections, minimize the inconveniences of manual corrections, and improve the resource and cost optimization involved in enhancing web accessibility. Such a tool will enable users to easily pinpoint accessibility errors and their corrections, effectively addressing the widespread issue of web accessibility errors for individuals who rely on ATs. In a commercial setting, such an approach can decrease annual web accessibility lawsuits raised in federal court.

This paper is organized as follows. Section 2 presents the creation of our dataset benchmark system for evaluating error correction. Section 3 details the role of ACCESS Agent in implementing corrections in the Document Object Model (DOM). Section 4 provides the results from prompt engineering approaches. Section 5 and 6 summarize our results and explores their implications for future work.

## 2 ACCESS BENCHMARK

To evaluate the success of our models, we created the first general automatic accessibility violation correction benchmark. In order to numerically measure the severity of violations, we follow regulation standards and map certain errors to one of five impact categories: critical, serious, moderate, minor, and cosmetic. Additionally, we introduce a numerical weighting to each category and calculate an aggregated severity score, which is used to evaluate the performance of a correction-proposal model. The initial severity score of the benchmark was 614, with an average of 24.56 for our dataset of 25 URLs.

### 2.1 DATA COLLECTION

We used the Playwright API to perform tests on website URLs and acquire web accessibility violation errors (e.g. empty headings, image tags without alternate text, and website links without

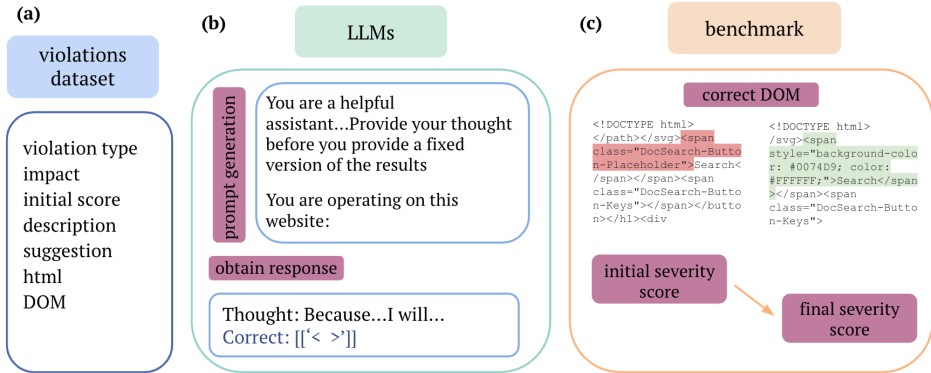

Figure 3: Overview of the violation correction approach. **(a)** The violations dataset includes columns extracted from Playwright accessibility tests, used for prompt generation. **(b)** We generate the prompt using prompt engineering techniques and traverse the dataset to obtain corrected HTML tags from LLMs. **(c)** Our benchmark compares initial and final severity scores of errors in the DOM.

Table 1: Description of dataset variables. Additional columns were included in our dataset but information was only extracted or modified from these main columns.

| Name | Description | Type |
|---|---|---|
| webURL | website URL passed into dataframe | text |
| numViolations | number of violations from URL | integer |
| id | violation type | text |
| initialScore | sum of numerical values of impact | integer |
| description | purpose of accessibility feature | text |
| help | describes accessibility requirements | text |
| html | HTML elements from violation nodes | HTML tags |
| DOM | gives DOM structure of URL | text |
| DOMCorrected | DOM structure with errors corrected | text |

discernible text). Playwright tests return a string of information relating to all the accessibility violations of a given URL. We generated a dataframe that organized violations for every website for approximately 25 URLs, which resulted in 171 rows of violations. Our data set includes website URLs hand-selected from research on commonly used, accessible, non-accessible, and older websites. Some example websites included were Google Calendar, Slack, Audible, BBC, Quora, New York Times, and ResearchGate. This dataset is valuable for the research community for various applications, such as predicting web accessibility failures, categorizing accessible websites, developing adaptive methodologies, and evaluating web accessibility improvements over time (Acosta-Vargas et al., 2020).

We extracted HTML and failure summary information from Playwright tests to include as columns in our data set. A DOM column was created to access the logical structure of the webpages and modify the HTML code. Analyzing our dataset, approximately 40% of URLs exhibited fewer than 5 violations and 60% exhibited 5 or more violations.

To generate the severity scores, a 1 was assigned to each "cosmetic" error (as specified by Playwright), 2 for "minor," and up to 5 for "critical." Each URL was assigned an initial severity score that was calculated by summing the total severity scores of all individual errors for the URL.

## 3 ACCESS AGENT

ACCESS Agent produces corrected DOM for websites in the dataset by feeding an LLM model with the error, description, suggested change, and incorrect HTML tag. The LLM returns a corrected

HTML tag and ACCESS Agent implements the correction process by inserting corrected elements into the DOM.

## 3.1 REACT PROMPTING

To obtain web accessibility violation corrections from our models, we first leveraged ReAct prompting with the action space being a 2D array of the corrected elements. By accessing incorrect HTML tags from our dataset, we built the prompt to feed to our transformer models. We provided a system message that served as a guideline for the model, presenting clear references to the identified incorrect HTML, error type, error description, suggested change, and the desired response format. This structured approach enabled the model to focus on specific accessibility issues and their respective corrections. To implement the ReAct prompting, we included an example response in our system message that structured the thought. This thought process acts as a validation mechanism, confirming that the model understands the task and the context of the accessibility violation. Our prompting system ensures that every response returned by our models includes reasoning for the correction and a corrected HTML tag.

```
System Message:
You are a helpful assistant who
will correct accessibility
issues of a provided website.

Provide your thought before you
provide a fixed version of the
results.

E.g.
Incorrect: [['<h3></h3>',
'<h3></h3>', <h3></h3>']]
Thought: because ... I will ...
Correct: [['<h3><Some heading
text</h3>', '<h3><Some heading
text</h3>', '<h3><Some heading
text</h3>']]
```

```
User Message:
You are operating on this
website: www.playwright.dev

Error: color-contrast
Description: Ensures the
contrast between foreground
and background meets WCAG 2 AA
minimum contrast ratio
thresholds
Suggested change: Elements
must meet minimum color
contrast ratio thresholds
Incorrect: [['Search']]
```

Figure 4: Example ReAct prompt messages for a color contrast error. Additional example errors were included in the system message for other prompting techniques.

## 3.2 FEW-SHOT GUIDED PROMPTING

We implemented Few-Shot guided prompting by providing more violation-correction examples in our system message. We utilized four example input and output messages, clearly labeled like the ReAct prompts. However, the Few-Shot prompt included only the incorrect and correct lines of HTML and did not include a thought. The examples included a missing header, a missing image description, a missing href URL, and a missing landmark as violation errors. This prompting style enables the model to better learn in context and prepare for subsequent examples in which a response is desired.

## 3.3 TRANSFORMER MODELS

Based on the prompt engineering techniques presented above, a system was created to generate the system and user messages for the desired prompt format, which could be used to generate prompts for correcting any individual error. We generated the system and user messages by accessing specific columns of the dataset for each error and formatting the information through prompt engineering (see Figure 4 for an example). These generated prompts were fed into our model. RegEx was used to search for the correction returned by the GPT response and return the correct HTML header.

To implement all the corrections on the entire dataset, we iterated through groups of the same URL, modifying the DOM by finding the incorrect HTML and replacing it with the corrected HTML returned by our model. By the end of the correction process, a new column in our dataframe was created, storing the corrected DOM for each URL. This process was repeated to test other large language models and prompt engineering techniques.

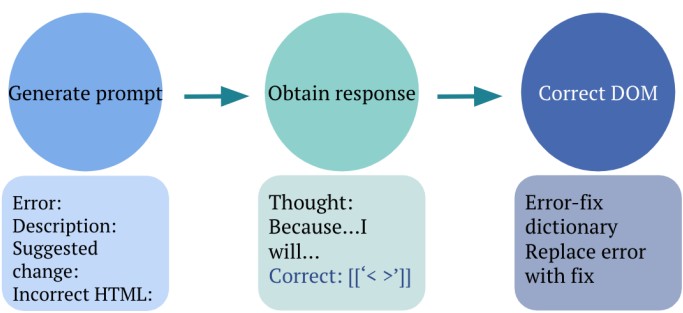

Figure 5: The DOM correction process using prompt engineering and LLMs.

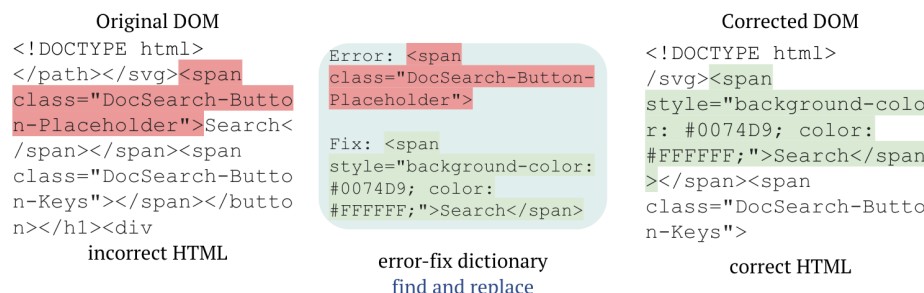

Figure 6: Example DOM correction process for one individual violation. This process was repeated for all violations in the dataset.

## 3.4 ACTION SPACES

Our goal is to demonstrate the efficacy of LLMs at correcting website accessibility violations in the DOM. This section describes how we implement HTML corrections and compute the improvement in violations.

We define $\mathcal{S}$ to be the set of tuples comprising error type, description of error, suggested change, and incorrect HTML tag. $\mathcal{M}_{\text{LLM}}$ is the LLM which takes $\mathcal{S}$ in the form of a user message and system message. $\mathcal{M}_{\text{LLM}}$ outputs a thought along with a corrected HTML tag.

We define the set of incorrect HTML tags obtained from Playwright tests as $PW(\mathcal{S})$.

To implement the corrections obtained from $\mathcal{M}_{\text{LLM}}(\text{PW}(\mathcal{S}))$, we substitute the incorrect HTML tag with the correction generated from $\mathcal{M}_{\text{LLM}}$. Let $\mathcal{F}_{\text{HTML\_fix}}$ represent the set of HTML code in which the incorrect HTML is replaced with the correction generated by $\mathcal{M}_{\text{LLM}}$.

$$\mathcal{F}_{\text{HTML\_fix}} = \{\text{sub}(\text{PW}(\mathcal{S}), \mathcal{M}_{\text{LLM}}(\text{PW}(\mathcal{S})))\}$$

We then compare the impact scores before and after LLM corrections. For the dataset, we compute $\mathcal{R}_{\text{initial}}$ and $\mathcal{R}_{\text{fix}}$, the average impact score of all URLs in the violations dataset, where $n$ is the number of violations per URL, $m$ is the total number of URLs in the dataset.

$$\mathcal{R} = \frac{1}{m} \sum_{j=1}^{m} \sum_{i=1}^{n} \text{impact}(\mathcal{S})$$

Finally, we evaluate the results by computing $\mathcal{I}$, the percentage average decrease in the number of violations after implementing $\mathcal{F}_{\text{HTML\_fix}}$.

$$\mathcal{I} = 1 - \frac{\mathcal{R}_{\text{fix}}}{\mathcal{R}_{\text{initial}}} \times 100\%$$

## 4    RESULTS

After receiving corrected DOMs for the URLs and violations in our dataset, we ran the modified DOM through Playwright again to test for remaining violations. A severity score was calculated again after the DOM corrections, and we compared scores before and after the corrections to evaluate the success of our models.

The results are shown in table 2 below.

Table 2: Initial and Final Violation Severity Score Results

| Model | Prompt | Initial / Avg | Final / Avg | % Score Decrease |
|-------|--------|---------------|-------------|------------------|
| GPT-3.5-turbo-16K | Few-Shot Guidance | 614 / 24.56 | 372 / 14.88 | 39.414% |
| GPT-3.5-turbo-16K | Chain of Thought | 614 / 24.56 | 334 / 13.36 | 45.603% |
| GPT-3.5-turbo-16K | ReAct | 614 / 24.56 | 299 / 11.96 | 51.303% |

Examining Table 2, our prompt engineering approach was able to successfully decrease severity scores for all three prompting methods using the GPT-3.5-turbo-16K models. A notable observation is the performance of ReAct prompting, which can be attributed to its effectiveness in overcoming the hallucinations that chain-of-thought prompting commonly suffers from. Chain-of-thought prompting requires the model to generate its own internal representations of knowledge, which can lead to hallucinations without a verification system like ReAct prompting or the ability to guide the model in a specific direction by pre-feeding it a thought (**?**). In comparison to other prompt engineering techniques, ReAct prompting's superior performance stems from its ability to blend reasoning with action, utilizing in-context examples to adapt to different environments.

Furthermore, in a baseline test involving a dataset of 10 URLs, we applied the same methodology and found that the GPT-4 model outperformed the GPT-3.5-turbo 16K by 4.891% in ReAct prompting. The enhanced performance of GPT-4 is likely due to its training on a more extensive dataset and its improved capability in processing complex concepts. However, testing our larger dataset of 25 URLs was limited by GPT-4's smaller token limit. Table 3 details the categorization of websites included in our dataset. Labels were assigned by hand; the distribution illustrates a range of websites that can be impacted by accessibility issues, which were represented in our dataset.

These findings underscore the LLM's capability to reduce accessibility violation errors through DOM correction. As detailed in Table 4, the model demonstrated strong proficiency in addressing text-based web accessibility issues, achieving 100% success rates in correcting violations of landmark-no-duplicate-content, label, skip-link, and ARIA-required-attribute. These violations, more straightforward text modifications or matching, are more easily processed by the model compared to those needing a comprehensive understanding of visual and layout aspects, such as meta-viewport and region. The model's moderate success ($> 60\%$) in correcting visual violations such as color-contrast, meta-viewport, and region, however, indicates some capability in handling responsive design elements, albeit with limitations. The lower success rate of the region violation reflects the complexity involved in discerning the semantic structure of web pages, a task demanding an in-depth understanding of both content organization and ARIA landmark roles. ARIA, or Accessible Rich Internet Applications, is a suite of attributes and roles designed to enhance the semantic value of web elements and complex interfaces, particularly for assistive technologies, by conveying states, properties, and roles beyond what's inherently available in HTML. Consequently, ARIA errors, which are especially prevalent in compound components and nested elements, often originate from different parts of the code, making automated correction a challenge. This complexity necessitates a nuanced understanding of the entire DOM, which the GPT Model, limited to correcting only one line of incorrect HTML, currently lacks. Addressing ARIA errors effectively may require a more

Table 3: Percentage of Types of Websites Within Dataset.

| Category | Percentage |
|---|---|
| Shopping/Transaction | 40% |
| Communication/Social | 20% |
| Government/Important | 4% |
| Research | 16% |
| Organization | 4% |
| News/Information | 16% |

Table 4: Most Frequently Corrected Errors After Prompt Engineering.

| Error ID | Percentage Corrected |
|---|---|
| landmark content | 100% |
| label | 100% |
| skip link | 100% |
| ARIA required attribute | 100% |
| landmark one main | 86% |
| linked name | 78% |
| color contrast | 75% |
| HTML has lang | 75% |
| meta viewport | 67% |
| region | 62% |

comprehensive approach, possibly extending the input of violations from one line to the entire DOM or incorporating other technologies, such as GPT-4 Vision to address a wider range of accessibility issues, including those based on visual elements.

Table 5: Most Common Violations within Dataset. The most common error, region errors, refer to content inappropriately contained by landmarks

| Error Type | Percentage |
|---|---|
| region | 12.28% |
| color contrast | 7.02% |
| landmark-unique | 6.43% |
| heading-order | 5.85% |
| duplicate-id | 5.26% |
| link-name | 5.26% |
| image-alt | 4.68% |
| landmark-one-main | 4.09% |

Additionally, the prevalence of non-text-based violations, such as region and color-contrast, which make up 12.28% and 7.02% of our data set respectively (Table 5), highlights the limitations of current LLMs. This insight is vital for guiding future research and development, suggesting areas where LLMs could be improved or combined with other technologies, like GPT-4 Vision, to address a wider range of accessibility issues, including those based on visual elements. Future work, as indicated by Table 3, could also focus on datasets primarily consisting of Government/Important, Research, and News/Information websites, as these categories typically have higher numbers of text-based violations and to test what specific text-based violations the model performs better in.

## 5 DISCUSSION

With the ubiquitous influence of the internet and the increasing societal reliance on online tools, ensuring unimpeded access to web content for all users, irrespective of their physical or cognitive challenges, stands as a paramount concern (Illingworth, 2001),(Pascual-Almenara et al., 2015). Our research pioneers an innovative automated approach using LLMs for efficient, low-cost violation corrections. To our knowledge, existing approaches for correcting accessibility violations do not automate the correction process using prompt engineering techniques in a way that can be easily adapted as a tool for users. It is imperative to acknowledge the broader societal ramifications of such results: an absence of comprehensive web accessibility may prevent individuals with impair-

ments from utilizing indispensable online services, notably services such as healthcare, banking, or document signing.

Our methodology involves correcting HTML tags of the Document Object Model. Web accessibility violations were systematically identified from a spectrum of websites via the Playwright API. This process yielded a comprehensive dataset composed of detailed violation specifications and the associated DOM. Following data acquisition, ACCESS Agent implements the correction automation process, utilizing OpenAI's GPT models. Through prompt engineering techniques, the model was fed violations, and revised HTML code suggestions were extracted to remediate the violations in each website's respective DOM. ACCESS Benchmark is our proposed evaluation schema, which compares summed numerical representations of the severity of website violations before and after correction to quantify the effectiveness of the automated process.

Our findings provide compelling evidence of the existing gaps in website accessibility corrections and the potential of our proposed solutions to address them. In the context of these results, it becomes evident that the ever-increasing reliance on the internet and websites for communication, commerce, and acquisition of knowledge accentuates the urgency of implementing comprehensive and effective web accessibility solutions. Accessibility barriers not only inhibit user experience but potentially ostracize a subset of the population from engaging fully in a digital society (Kurt, 2018),(Abu Addous et al., 2016). This is particularly consequential in critical domains such as healthcare, where web inaccessibility can translate into a denial of vital services (Sarita et al., 2021). Therefore, this research not only holds significance beyond the realms of digital design and Natural Language Processing, but also promotes broader dialogue on societal inclusivity and equality in the digital era. It underscores the importance of our project as a practical tool for end-users, emphasizing its role in facilitating more accessible and equitable digital environments.

## 6 CONCLUSION

Our research efficiently identifies and corrects web accessibility violations according to the Web Content Accessibility Guidelines (WCAG) using OpenAI's GPT models and prompt engineering techniques. The outcome was a marked decrease in severity scores, dropping by 51% using the GPT 3.5-turbo-16K model. While our results still leave accessibility violations to be corrected, it is notable that an automated approach was able to identify and correct HTML in the DOM without the need for manual individual error correction. Ultimately, our results are a major step toward digital inclusivity, inventing a new sub-field of automated accessibility that leaves more to be discovered.

Moreover, the real-world implications of our study are significant. Such automated violation correction methods could be adapted into a web-based tool or extension to enable users to correct accessibility violations of websites when needed. This research lays the groundwork for a user-friendly and robust correction approach, which not only benefits web developers and companies but also significantly enhances the online experience for people with disabilities. The widespread adoption of such a tool could lead to a host of advantages: enhanced digital accessibility, increased business revenue through more inclusive websites, reduced legal issues related to website accessibility, lower maintenance costs, and, most importantly, the creation of a truly inclusive online environment for everyone.

Looking forward, future efforts could approach augmenting the diversity of our violations dataset. Expanding the dataset to encompass websites from varied industries, sectors, and geographic regions would empower our models to effectively address an even broader spectrum of accessibility challenges. The integration of deep learning techniques, such as coupling computer vision with natural language processing, could introduce the ability to detect and rectify issues spanning multimedia elements and textual content, further elevating the comprehensiveness of accessibility enhancement.

## 7 REPRODUCIBILITY STATEMENT

Supplementary materials include the violations dataset generated by Playwright, the code used to run ACCESS benchmark and ACCESS agent, and the dataset with corrected DOM elements. Results may vary when running the models as the responses returned from GPT models are not necessarily replicable.

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
