# OpenReview forum: "Deep-Learning Approaches for Optimized Web Accessibility: Correcting Violations and Enhancing User Experience"
_ICLR.cc/2024/Conference — Submitted to ICLR 2024_

### Official Review · Reviewer_72AK · 2023-10-29

**Soundness:** 2 fair
**Presentation:** 2 fair
**Contribution:** 2 fair
**Rating:** 3
**Confidence:** 4

**Summary:**

The paper introduces an approach to automatically correct accessibility issues on websites, aiming to improve
accessibility for individuals with impairments while minimizing the need for manual corrections. To achieve this, the
authors utilized an established website accessibility evaluation tool to identify potential issues within the HTML code.
This process yielded a dataset containing the violations of 25 URLs. By leveraging OpenAI's GPT models and prompt
engineering, the system automatically revised the HTML code, resulting in a notable decrease of the accessibility
violation errors.

**Strengths:**

The issue of accessibility is highly relevant and research in this area is definitely required to advance automation.
The presented system proved to be able to automatically improve HTML code to successfully increase accessibility. The
evaluation, although not particularly extensive, is transparent and traceable. Since the authors have provided the dataset
and the code, it should be possible to reproduce the results to a certain extent. To enable future comparisons with other
methods, the resulting dataset should be made publicly available.
The authors have cleverly used existing components, such as OpenAI's GPT model, and creatively combined them to
solve an essential and socially relevant problem.
The approach certainly has potential and quality, but the paper needs some revision, as it does not seem quite mature in
its current form.

**Weaknesses:**

Although the introduction effectively underscores the importance of enhancing and automating accessibility, it could benefit from a more comprehensive integration within the landscape of the really closely related work. While the authors touch on recent automatic and semi-automatic approaches, they just mention that there have also been studies that considered improving accuracy through semi-automatic techniques of manual correction and even a few automating this correction using prompt engineering (most related), without providing specific examples or literature. If the deployment of prompt engineering is something completly new in this context, the authors must either point this out more clearly or highlight the differences and advantages of their approach compared to the existing state of the art.
Regarding the cited related work, it should also be noted that the mentioned automatic strategies mainly cover the generation of alternative text for images and in this context the insufficient quality of the automatic alt-text approach of facebook is pointed out. This aligns with the statement on page 1, that the most common issue in accessibility is the lack of text equivalents or alternate text for images, but is also incidental after it becomes apparent (page 7) that images cannot be processed at all by the system presented by the authors.

With regard to the proposed method, it is difficult to tell which components were pre-existing and which were developed by the authors, making the contribution not directly apparent. The presented approach does not seem to involve any fundamentally novel non-basic components, but its strength lies in the adept engineering and smart combination of existing components. This has certainly its own value when it is shown that an existing problem can be solved in new ways. Hence this is not really the issue.
However, my main criticism is that there is no quantitative or qualitative comparison with existing methods.
As much as I appreciate the research in this area and recognize the need for automated methods, there is unfortunately no substantial evidence presented that the authors' method is in any way at least equivalent to existing approaches, or even superior to them.
The authors should try to compare with existing approaches, as the mere improvement over the initial situation is arguably not so meaningful.
This would be different, if their method uniquely addresses unresolved accessibility issues and a comparison might not be feasible or possible. However, it appears that this is not the case, or at least it is not evident and must be clearly pointed out and proven by facts.
The same applies to the authors' claim within the conclusion section that they have invented a new subfield of automated accessibility.

Other weaknesses:
- I don't see the necessity of the mathematical formulas in section 3.4. The expressions appear to artificially introduce complexity, making it more unnecessary challenging for the reader to comprehend. Rather than introducing variables with no subsequent significance, it would be more effective to use an illustration of the pipeline or at least pseudo code. Especially since the definitions lack clarity and precision, for instance, the term "sub" was used without prior explanation, and both the definition and the necessity of F_{HTML\_fix}'s definition remain incomprehensible, raising doubts that it is mathematically well-defined. Furthermore, most of the variables are unnecessary for defining the terms of R and I. A clear verbal introduction would have sufficed and enhanced comprehension.
- Section 5 is not really a discussion, but a summary. There is no critical examination present.

Minor weaknesses:
- The title is misleading. "Deep Learning" refers solely to the use of pretrained LLMs and not to the direct embedding of deep learning techniques.
- In table 2 the space is missing in some cells of "final / avg".
- Typo in Figure 4: "messasges"
- Page 2 and 3 contain faulty links to literature
- The figures/tables should be referenced in the main text to put them in context and improve readability by guiding the reader. Also, the context of some tables, such as Table 3 , is not discussed or mentioned in any way, so it is not clear what the reader is supposed to do with this information.

**Questions:**

- Why is there no comparison to other existing methods?
- Since missing alternate text for images is in general a key issue, how would approaches addressing alt-text, such as the
one of Facebook, perform on your data set? Even if the authors' method ignores images, this could help to assess the
overall performance (assuming the URLs contain a realistic number of images).
- How were the 25 URLs in the dataset selected? Which URLs were previously used by related work? Why weren't the
same ones used, or is there no information available?

---

> ### Author Response · Authors · 2023-11-23
> **Response to Reviewer 72AK Part 1/2**
>
> We thank the reviewer for their thorough feedback and thoughtful comments.
>
> **Changes made to the pdf:**
> * introduction has been improved to include better context among relevant work. Notably, we emphasize the novelty of our approach by framing it as a tool for end users vs. companies or web developers, hence the difficulty in comparing our results to existing research.
>     * (Section 1) “Currently, no accessibility correction tool exists with the intent of serving users with a tool to correct web pages on the go. While the scope of previous studies was limited to addressing alt-text generation with computer vision, our tool can be used more ubiquitously and purposefully by web users to handle web accessibility violations when desired.”
> * > *no quantitative or qualitative comparison to existing methods*
> >
>     * throughout the paper, we have updated the introduction, results, discussion, and conclusion to better incorporate references to existing work and to support the novelty of our work as a tool that can enable users to make websites more accessible as and when needed. To our knowledge, existing methods like ours (employing prompt engineering for automated text-based corrections accessible to users) are not in existence or feasible to compare to
> * math: the purpose of introducing mathematical expressions in section 3.4 is to provide a precise and formal representation of our approach. However, we understand the concern about the potential complexity for the reader. Before introducing the term “sub,” we clarify that “we substitute the incorrect HTML tag with the correction generated.” This explanation logically leads the reader to understand that “sub” serves to substitute an incorrect HTML for the correction. Our goal is to enable readers to grasp the intricacies of our approach without unnecessary complexity. Given the presence of Figure 5 and 6, which illustrate the correction pipeline with visuals, we believe that the combination of figures and mathematical expressions strike a balance between formality and accessibility in understanding.
>
> * > *Section 5 is not really a discussion, but a summary.*
> >
> The discussion has been updated to include more critical examination of the impact of our research and its significance within a real-world context. (Section 5) “To our knowledge, existing approaches for correcting accessibility violations do not automate the correction process using prompt engineering techniques in a way that can be easily adapted as a tool for users.”
> * missing spaces in Table 2 have been corrected, typo in Figure 4 corrected, citations have been corrected
> * all figures and tables are referenced in the main text and further discussed to explain their relevance in the context
>      * (Section 4) “As detailed in Table 4, the model demonstrated strong proficiency in addressing text-based web accessibility issues, achieving 100\% success rates in correcting violations of landmark-no-duplicate-content, label, skip-link, and ARIA-required-attribute”
> * >dataset should be made publicly available
> >
> The dataset is publicly available through GitHub.
> * > *Why is there no comparison to other existing methods?*
> >
> The absence of comparison to existing methods stems from the fact that there are currently no known benchmarks or papers that attempt to do exactly what we are doing, namely employing prompt engineering to correct accessibility violations in an efficient manner. As a result, there is no means to determine whether we are beating the state of the art. Instead of seeing our research as a tool to help developers quickly build accessible tools, we have reframed the practical use case of our approach to be something that an end user can use (e.g. via a Chrome extension) that automatically makes webpages accessible. This therefore avoids a comparison with humans in terms of accuracy when fixing the DOM.
>     * highlight: (Section 6) “Such automated violation correction methods could be adapted into a web-based tool or extension to enable users to correct accessibility violations of websites when needed.”

---

> > ### Author Response · Authors · 2023-11-23
> > **Response to Reviewer 72AK Part 2/2**
> >
> > > *Since missing alternate text for images is in general a key issue, how would approaches addressing alt-text, such as the one of Facebook, perform on your data set? This could help assess the overall performance (assuming the URLs contain a realistic number of images)*
> > >
> > For missing alt-text errors, our approach would likely be able to correct some but not all errors by suggesting possible alt-text using text-based context surrounding the presence of an image. Unlike research that might target individual issues like alt-text, our method employs a holistic strategy to comprehensively correct a wider range of violations present in web content. Our goal is to provide a more versatile and automated solution that can handle a diverse range of accessibility issues.
> >
> > Approaches specifically addressing missing alt-text may be able to rectify alt-text issues, but without a comprehensive view of other errors, the improvement in accessibility would still be limited. According to our dataset, there were still at least 6 types of errors that occurred more frequently than alt-text errors in the dataset, and employing an alt-text correcting approach on our dataset would still leave more than 95% of the accessibility errors unresolved.
> > > *How were the 25 URLS in the dataset selected? Which URLs were previously used by related work? Why weren’t the same ones used, or is there no information available?*
> > >
> > Currently, there is no available information on URLs employed in related work. In fact, to our knowledge, no research of similar approach exists in which the researchers test a collection of URLs and make the selected URLs publicly available. Playwright can only perform tests on URLs beginning with “http.” Hence, we hand-selected 25 URLs in an effort to diversify represented categories (see Table 3). In our new PDF, we have also added a line in Section 2.1 detailing some example URLs: “Section 2.1 “Some example websites included were Google Calendar, Slack, Audible, BBC, Quora, New York Times, and ResearchGate.” Future research would involve more efforts to incorporate a larger range of URLs to examine our approach as a tool in multiple contexts.

---

### Official Review · Reviewer_Q9fB · 2023-11-01

**Soundness:** 3 good
**Presentation:** 3 good
**Contribution:** 3 good
**Rating:** 5
**Confidence:** 3

**Summary:**

This paper presents deep-learning based technique to correcting accessibility violations in the web content. Their main goal is to promote web accessibility and inclusivity, and improve the overall user experience for people with impairments. They find issues in HTML code using website accessibility violation data and prompt engineering. They could get above 50% reduction in accessibility violation errors after corrections using accessibility error information, large language models and prompt engineering techniques. This paper shows an approach for inclusive web content, and advances to automate web accessibility.

**Strengths:**

1/ By automating accessibility violation corrections, the paper tends to eliminate the inefficiency and inaccuracy associated with manual error corrections.
2/ Their approach is able to decrease severity scores by over 50%
3/ They pioneer an innovative automated approach using LLMs for efficient, low-cost violation corrections.
4/ The paper promotes broader dialogue on societal inclusivity and equality in the digital era.

**Weaknesses:**

1/ Typo in line "Section 2 presents our the creation of our dataset benchmark system for evaluating error correction."
2/ Results from Table 2 to 5 are not discussed. Since the paper can be organised in a compact way, authors can discuss the result to give more clarity.
3/ FEW-SHOT GUIDED PROMPTING could be written more elaborately. Some of the questions: what does "by increasing the examples provided in the system message"mean?

**Questions:**

as above in weaknesses - major concern on discussion of results.

**Details Of Ethics Concerns:**

No concerns

---

> ### Author Response · Authors · 2023-11-23
> **Response to Reviewer Q9fB**
>
> We thank the reviewer for their feedback and for acknowledging our innovative approach and its impact on digital inclusivity.
>
> In our new pdf, we have addressed the following weaknesses to provide a stronger and clearer interpretation of our results:
> * typo in Section 2 resolved
> * more in-depth discussion of results from Tables 2 through 5 and their significance has been included in the paper
>     * Discussed the connection between the most frequent errors (visual/layout-based errors) and the model’s capability to correct text-based errors more easily than more semantic or visual errors
>     * (section 4) “As detailed in Table 4, the model demonstrated strong proficiency in addressing text-based web accessibility issues, achieving 100% success rates in correcting violations of landmark-no-duplicate-content, label, skip-link, and ARIA-required-attribute”
>     * (section 4) “Future work, as indicated by Table 3, could also focus on datasets primarily consisting of Government/Important, Research, and News/Information websites, as these categories typically have higher numbers of text-based violations and to test what specific text-based violations the model performs better in.”
> * discussion of results has been improved to add more clarity
> * Few-Shot Guided Prompting (Section 3.2) has been described in more detail. “the Few-Shot prompt included only the incorrect and correct lines of HTML and did not include a thought. The examples included a missing header, a missing image description, a missing href URL, and a missing landmark as violation errors.”
>
> >*What does “by increasing the examples provided in the system message” mean?*
> >
>  “Increasing the examples provided in the system message” refers to including more than one example error in the system message. For example, Figure 4 illustrates a system message with one example of a missing heading text error. To improve the performance of our model, we could experiment with including additional errors to provide more context. For example, we could add a color contrast error with an example correction, as well as a button name error with an example correction.
>
> To clarify this, we updated Section 3.2 with the explanation “We implemented Few-Shot guided prompting by providing more violation-correction examples in our system message.”

---

### Official Review · Reviewer_pEL2 · 2023-11-03

**Soundness:** 2 fair
**Presentation:** 2 fair
**Contribution:** 2 fair
**Rating:** 5
**Confidence:** 4

**Summary:**

The paper proposes an automated approach utilising the GPT-3.5-turbo-16K model with prompt engineering to fix over 50% of 171 web accessibility violations from 25 websites. By integrating the Playwright API for accessibility testing and leveraging the GPT-3.5-turbo-16K model for corrections, the claims that this would be a more efficient alternative to manually correcting errors.

**Strengths:**

The paper was generally easy to follow. The authors explored different prompting techniques and had a good discussion on the strengths and limitations of the GPT-3.5-turbo-16K model in fixing the accessibility issues.

**Weaknesses:**

Sec 3.3 was unclear: What system was created for generating the prompts? The types of websites in the study were also unclear, e.g., were these static/dynamic papers? The authors tested their approach with GPT3.5 and GPT4 only; a few other models could be explored to test the generalisability of the method.

There needs to be a baseline to compare the performance of the approach. The score allows us to compare the different prompting techniques but not judge whether the approach overall is better than humans with accessibility testing tools.

Minor typos/comments:
- A few references were missing -- pages 2 and 3
- table II below -- Table 2? (page 7)
- What is ARIA; define?

**Questions:**

1. What types of websites were used: dynamic or static? Were there React apps or similar apps? OR Would this approach work for React or similar apps?

2. Were there any perfect web pages? If yes, what did the proposed model do in that case?

3. What was the maximum number of violations observed?

4. Were the typos in the "Suggested change" part of Fig 4 intentional? Was this an actual example?

5. How many system and user messages were generated?

6. Can you explain the process that was used to generate the "suggested change" and "incorrect" in the User Message? Were these done manually, and if so, how did the team decide the best phrasing for the suggested changes?

---

> ### Author Response · Authors · 2023-11-23
> **Response to Reviewer pEL2**
>
> We thank the reviewer for their insightful questions and for appreciating our clear discussion of prompting techniques and models.
>
> In this comment, we respond to the reviewer’s questions and outline key changes made to the pdf.
>
> >*1) What types of websites were used: dynamic or static? Were there React apps or similar apps? OR Would this approach work for React or similar apps?*
> >
> The types of websites used for our benchmark were all dynamic. For example, we included websites such as New York Times, Audible, and Google Calendar. However, the websites were fed through Playwright based on their state at the time of dataset creation.
>
> In the new PDF, we listed some example websites in our dataset in Section 2.1 (“Some example websites included were Google Calendar, Slack, Audible, BBC, Quora, New York Times, and ResearchGate.”)
>
> There were some websites that were built using React, including Collegeboard, SoundCloud, Vimeo, BBC, Eventbrite, GrubHub, New York Times, Patreon, and ResearchGate. This approach worked for these websites, and they were able to correct several of their violations.
>
> >*2) Were there any perfect web pages? If yes, what did the proposed model do in that case?*
> >
> There were no web pages within our dataset that had zero accessibility errors. This is due to the fact that when run through Playwright, a web page without any violation errors would not return any results and therefore could not fill up dataset columns. However, based on some tests conducted during our research, if a web page has zero violations, the LLM may return a “correction” that results in the same initial HTML tag (and thus if the replacement process were applied using the LLM’s response, there would be no change in the DOM).
>
> >*3) What was the maximum number of violations observed?*
> >
> In our dataset, the maximum number of violations initially observed was 13. It is likely that this number could be much larger if we expanded our dataset to include more URLs.
>
> >*5) How many system and user messages were generated?*
> >
> Each violation generated one system message and one user message. So since we had 171 total rows in our dataset, 171 system messages and 171 user messages were generated.
>
> >*6) Can you explain the process that was used to generate the “suggested change” and “incorrect” in the User Message? Were these done manually, and if so, how did the team decide the best phrasing for the suggested changes?*
> >
> The “suggested change” and “incorrect” information in the user message was extracted from the dataset. The “incorrect” comes from the “html” column of the dataset and “suggested change” comes from the “help” column of the dataset. This information was returned by Playwright for each website and formatted into our dataset to be used for the user messages.
>
> **Changes made to the pdf**
> * References have been corrected and updated
> * Table 2 reference in Section 4 has been updated as “table 2”
> * Section 4 includes a brief definition of ARIA: “ARIA, or Accessible Rich Internet Applications, is a suite of attributes and roles designed to enhance the semantic value of web elements and complex interfaces, particularly for assistive technologies, by conveying states, properties, and roles beyond what's inherently available in HTML.”
> * Typos in Fig 4 have been updated. This is an example of a system message and user message that would be fed into the models.
> * Section 3.3 has been updated to include more detail on the process of generating the prompts. “We generated the system and user messages by accessing specific columns of the dataset for each error and formatting the information through prompt engineering (see Figure 4 for an example).”
>
> Overall, we reframed our approach to be more of a tool for individual end users, instead of for web developers or companies. As a result, a comparison to existing methods is unfeasible, but the ability to decrease accessibility violations is notable for serving users with impairments.

---

> > ### Comment · Reviewer_pEL2 · 2023-11-23
> >
> > Thanks for your clarifications. I have read your responses. I kept my score.

---

### Public Comment · ~Jonathan_Robert_Pool1 · 2025-06-08

The paper says “Playwright tests return a string of information relating to all the accessibility violations of a given URL.”

That is not correct. Playwright tests are whatever tests the Playwright user uses Playwright to perform. If the paper specifies what tests were performed, I have not found that information here.

The supplementary materials show that axe-core 4.8 is used.

---

### Meta-Review · Area_Chair_bv8k · 2023-12-10

**Metareview:**

The paper discusses an automated approach using the GPT-3.5-turbo-16K model with prompt engineering to address web accessibility violations. The goal is to enhance web accessibility and inclusivity by automatically correcting of violations in the HTML code, thus improving the user experience for individuals. The approach involves leveraging the Playwright API for accessibility testing and employing large language models to automate the correction process, presenting a more efficient alternative to manual error correction. In the experiments, the proposed approach  fix over 50% of 171 web accessibility violations from 25 URLs

Pros (from reviewers):
1. Highly motivated direction to leverage the LLM to automatically improve the web accessibility.
2. Clever way to utilizing existing LLM API for a new application.

Cons (from reviewers):
1. Baselines or more analysis are needed to better understand the performance of the approach. e.g. how much the improvement impact the actual user experience.
2. Some details of experimental setup are not clear, more in depth discussion are needed for the proposed components and results.

The authors actively participated in the rebuttal and provided responses on those weaknesses and questions. However, the major concerns remains after rebuttal.

**Justification For Why Not Higher Score:**

The scores indicate a clear rejection. 2 of the 3 reviewers responded that they have looked at the reviews but would like to keep the score.

**Justification For Why Not Lower Score:**

N/A

---

### Decision · Program_Chairs · 2024-01-16

Reject